# Human Triosephosphate Isomerase Is a Potential Target in Cancer Due to Commonly Occurring Post-Translational Modifications

**DOI:** 10.3390/molecules28166163

**Published:** 2023-08-21

**Authors:** Sergio Enríquez-Flores, Ignacio De la Mora-De la Mora, Itzhel García-Torres, Luis A. Flores-López, Yoalli Martínez-Pérez, Gabriel López-Velázquez

**Affiliations:** 1Laboratorio de Biomoléculas y Salud Infantil, Instituto Nacional de Pediatría, Secretaría de Salud, Mexico City 04530, Mexico; ignaciodelamora@yahoo.com.mx (I.D.l.M.-D.l.M.); garcia.itzhel@gmail.com (I.G.-T.); 2Laboratorio de Biomoléculas y Salud Infantil, CONAHCYT-Instituto Nacional de Pediatría, Mexico City 04530, Mexico; luisbiolexp@gmail.com; 3Instituto Tecnológico y de Estudios Superiores de Monterrey, Mexico City 14380, Mexico; yoalli.martinez@tec.mx

**Keywords:** glycolysis, deamidation, phosphorylation, S-nitrosylation, post-translational modification

## Abstract

Cancer involves a series of diseases where cellular growth is not controlled. Cancer is a leading cause of death worldwide, and the burden of cancer incidence and mortality is rapidly growing, mainly in developing countries. Many drugs are currently used, from chemotherapeutic agents to immunotherapy, among others, along with organ transplantation. Treatments can cause severe side effects, including remission and progression of the disease with serious consequences. Increased glycolytic activity is characteristic of cancer cells. Triosephosphate isomerase is essential for net ATP production in the glycolytic pathway. Notably, some post-translational events have been described that occur in human triosephosphate isomerase in which functional and structural alterations are provoked. This is considered a window of opportunity, given the differences that may exist between cancer cells and their counterpart in normal cells concerning the glycolytic enzymes. Here, we provide elements that bring out the potential of triosephosphate isomerase, under post-translational modifications, to be considered an efficacious target for treating cancer.

## 1. Introduction

Cancer is a significant cause of death globally, with almost 10 million deaths and 20 million new cases reported in 2020. While cancer is more common in older individuals, it can affect people of all ages, including children. In fact, around 400,000 children were diagnosed with cancer in 2019 [1]. Among deaths of young individuals (0–19 years old), cancer accounted for 2.54% of all deaths, while it represented 22.61% in older age groups (20 to >85 years old) [2]. When comparing cancer incidence by age group to other diseases, it accounted for 1.07% in young people and 12% in older ages during the same period (Figure 1) [1,2].

Cancer is a term used to describe a vast group of diseases where abnormal cells grow uncontrollably in any part of the body. They can exceed their usual boundaries and invade neighboring areas while also spreading to other organs. Neoplasms and malignant tumors are other commonly used terms (http://www.who.int/topics/cancer/es/, accessed on 14 July 2023). Early detection of cancer is crucial for successful treatment. Imaging tests, such as radiology, ultrasound, magnetic resonance spectroscopy, tomography, laser optical tomography, etc., are commonly utilized in cancer diagnosis [3]. The diagnosis of cancer heavily depends on molecular biomarkers that include DNA, RNA, microRNAs (miRNAs), long noncoding RNAs (lncRNAs), and proteins [4,5]. Various techniques, including fluorescent in situ hybridization (FISH) [6], polymerase chain reaction (PCR), immuno-PCR [7,8], single-cell genomics [9], flow cytometry [10], and enzyme-linked immunosorbent assay (ELISA) [11], are used to analyze molecular biomarkers for better and more accurate diagnosis of certain pathologies. These techniques are commonly utilized to provide suitable diagnosis for this group of diseases.

It is notable that, to be helpful, biomarkers must distinguish between people with cancer and those without it [12]. The critical biochemical problem in oncology is the definition of the lesions that differentiate the neoplastic from the normally growing cell. Additionally, biomarkers for cancer can be associated with the factors that cause a healthy cell to develop into a cancerous cell, the effects of being a tumor cell, or both.

The cancer cell is characterized by its ability to escape from homeostatic controls. It can metastasize, invade, and grow independently of host controls, whereas other cells must contribute to the welfare of the host. One of the best-known features of neoplastic cells is the variation from phenotypically nonfunctional to phenotypically excessively functional [13,14]. Indeed, cancer cells undergo extensive metabolic remodeling [13], in which protein synthesis dysregulation is critical for cellular transformation. Various studies highlight the upregulation of several proteins as characteristic of different types of cancer and propose their use as targets to cope with them [15,16,17,18,19,20,21].

Among the proteins pointed to be targets or biomarkers in cancer, those coming from or to regulate glycolysis have been extensively studied mainly due to their upregulated expression [22,23,24,25,26,27,28]. Among such proteins, human triosephosphate isomerase (HsTIM) is considered a cancer-related biomarker based on its overexpression in various cancer types [29,30,31,32,33,34,35,36,37,38,39,40,41,42,43,44,45]. This has caught the scientific community’s attention to go further in developing new strategies for the treatment of cancer.

## 2. Cancer Treatment

Treating cancer continues to be a difficult challenge, despite many advancements made over the course of several decades. The treatments administered to cancer patients depend on the stage in which it is detected, the type of cancer, its development, and its extension. There are five available treatment options: surgery, radiotherapy, chemotherapy, hormonal therapy, biological therapy (immunotherapy and gene therapy), and a combination of one or more. Since chemotherapy is the most common treatment, more than 100 drugs are currently used alone or in combination with other medications. Many of these exert their action on cells that are actively dividing. However, they maintain a restricted selectivity towards cancer cells; therefore, they also significantly damage normal cells, causing severe side effects such as fatigue, hair loss, anemia, gastrointestinal problems, malnutrition, pain, and infections. Side effects considerably decrease quality of life and can even lead to the death of patients [46].

Moreover, cancer cells often develop resistance, either acquired or inherited, to various chemotherapeutic treatments. This exacerbates the already challenging situation, as widely demonstrated [47,48]. For this reason, efforts in research are ongoing to create new compounds that can selectively eliminate cancer cells with greater effectiveness.

Although the glycolytic pathway is being continuously proposed as a target to treat cancer, it has never been successfully exploited clinically, particularly since glycolysis inhibitors were proven to have undesirable side effects and limited efficacy in humans [49]. Despite those setbacks, several studies have investigated anaerobic glycolysis as a therapeutic opportunity in cancer. Small-molecule inhibitors, such as STF-31 directed to glucose transporter 1 (GLUT1) or glutor, and BAY-876, and BAY-897 directed to GLUT1, GLUT2, GLUT3, and GLUT4, are being studied to inhibit glucose uptake. Further, hexokinase is assayed against a series of 2,6-disubstituted glucosamines and (E)-N’-(2,3,4-trihydroxybenzylidene) aryl hydrazide derivatives [50,51]. Since lactate dehydrogenase is essential for the Warburg effect (see below), it has also been explored as a therapeutic target with molecules such as GSK 2837808A [52,53]. Further, kinetic modeling and metabolic control analysis identified hexose-6-phosphate isomerase as a possible target in cancer [54]. This enzyme exerts high control in cancer glycolytic flux but not in normal cells. Some *N*-substituted 5-phosphate-D-arabinonamide derivatives as ST082230 and ST078079, are being studied to inhibit this enzyme in cancer cells [55].

The glycolytic HsTIM enzyme is often found to be overexpressed in different types of cancer and linked to metastatic phenotypes. As a result, researchers have investigated this protein as a potential target for treating these diseases [30,56]. Contrary to what happens in developing drugs directed to parasite TIMs [57,58,59], the main reason for failing to find a drug against HsTIM is that the structure of this enzyme is conserved between metastatic and normal cells.

In this regard, some of the most successful strategies to achieve selectivity toward cancer cells are therapies based on metabolic differences between the target (cancer) cells and normal cells. Therefore, this review focuses on the metabolic post-translational differences found in HsTIM from cancer cells and how they can be utilized. Indeed, there are very few review antecedents on this topic [56,60]. Therefore, it is very timely to analyze the scientific literature to show the opportunity to develop drug treatments targeted to triosephosphate isomerase regarding the common metabolic post-translational modifications (PTMs) that this enzyme can undergo during cancer.

## 3. Cancer Cell Metabolism

The German biochemist Otto Heinrich Warburg first described the link between tumorigenesis and deregulated metabolism a century ago [61,62]. He found that cancer cells favor glucose uptake and lactate production with dysfunctional mitochondria. This was called aerobic glycolysis or the Warburg effect [63]. Currently, we have learned that mitochondria remain functional in cancer cells [64]. Instead, the Warburg effect is an essential characteristic of metabolic reprogramming, resulting from the interplay between (normoxic/hypoxic) hypoxia-inducible factor-1α (HIF-1α) overexpression, oncogene activation (cMyc, Ras), loss of function of tumor suppressors (mutant p53, mutant phosphatase and tensin homolog (PTEN), microRNAs and sirtuins with suppressor functions), activated (PI3K–Akt–mTORC1, Ras–Raf–MEK–ERK–cMyc, Jak–Stat3) or deactivated (LKB1–AMPK) signaling pathways, components of the tumor microenvironment, and HIF-1α cooperation with epigenetic mechanisms [65,66].

The data mentioned above are a product of the recent advancements in the field of cancer metabolism over the past few decades. Such advances have identified at least six hallmarks of cancer acquired during multistep development: (1) increased uptake of glucose and amino acids from the tumor microenvironment and concomitant delivery of lactate and protons to tumor cells; (2) increase of the glycolysis pathway, the pentose phosphate pathway, and the tricarboxylic acid cycle intermediates to build and sustain the exacerbated proliferation of cancer cells; (3) a more mechanical uptake of nutrients via phagocytosis, invasion into another cell’s cytoplasm, and micro- and macro pinocytosis; (4) increased need and utility of nitrogen derivatives and their conversion to nucleotides, non-essential amino acids, and polyamines; (5) changes in metabolite-driven gene regulation by modifications such as methylation, acetylation, and succinylation; and (6) bilateral metabolic interaction through the exchange of nutrients and amino acids or the influence of growth factors or environmental conditions, such as hypoxia or redox stress on the microenvironment or their influence on metabolism and the different signaling levels of cancer cells [65,66]. These characteristics occur in varying degrees and circumstances of many types of cancer.

Metastatic cells reconfigure their metabolism to support oncogenic processes such as accelerated growth and proliferation. Growing evidence indicates a complex interaction between tumor metastasis and metabolic rewiring in cancer. Metabolic rewiring could drive metastasis by (1) generating oncometabolites to take control of metastatic signaling cascades via regulating gene expression, (2) generating metabolites/cofactors that act as agonists/antagonists for functional proteins involved in metastasis, and (3) modulating metabolic demands of cancer cells, thus allowing adaptation to the various stages of the metastatic cascade. On the other hand, metastatic-associated signaling can influence cellular metabolism by directly affecting the expression and activity of metabolic enzymes [67].

Depending on the tumor growing pattern, cancer cells preferentially use glucose or amino acids for their energetic or biosynthetic needs. Lipids, particularly fatty acids, can also be taken up by the tumor cell. The mechanisms by which this occurs likely involve alterations to genes that encode metabolic enzymes [68]. For example, glutamine metabolism is accelerated and essential for biomass production since it is the primary electron donor in the anaplerotic synthesis of aspartate. Aspartate is known to be critical to proliferating cells’ respiration, as occurs in cancer processes [69]. Another example is lipid metabolism, which is also increased since it is very effective for obtaining energy through β-oxidation. This has been evidenced in some types of cancer due to the increase in the transcription of genes related to this metabolic pathway, such as the CD36 receptor. This receptor initiates the signaling cascade to internalize lipids from the extracellular medium [70].

Further, cancer phenotypes with metabolic deficiencies have been reported as occurring in tumors such as hepatocarcinoma, leukemia, and melanoma. Such types of cancer do not express any of the enzymes necessary for arginine synthesis, turning them into auxotrophs for this amino acid. In this sense, enzymes such as arginine deaminase inhibit these tumors’ growth by limiting their substrate’s extracellular availability and have been used as an alternative therapy to combat them [71].

However, without diminishing the importance of the variety of metabolic changes in such diseases, those occurring in the glycolytic pathway are critical in initiating and progressing cancer. Hence, the overexpression of glycolytic enzymes is caused by active metabolic reprogramming required to support sustained cancer cell proliferation and malignant progression [72]. As already mentioned, such upregulation of the genes involved in glycolysis is part of the so-called Warburg effect, which is present in more than 70% of all human cancers. [72,73]. This is certainly worthy of special attention in current and future cancer research.

## 4. The Glycolytic Pathway Could Play a Crucial Role in the Energy Metabolism of Cancer Cells

As mentioned, many studies have described cancer cells as highly energy-demanding. The glycolytic pathway rapidly supplies such energetic demand since the speed of the cytosolic ATP generation is approximately 100-times faster than in mitochondria [72]. In glycolysis, a series of enzymatic events breaks down glucose to pyruvate. In this process, two ATP molecules and two NADH molecules are generated for each glucose molecule. In the so-called first phase of the pathway, glucose is phosphorylated, isomerized, and finally degraded to two three-carbon molecules, dihydroxyacetone phosphate (DHAP) and glyceraldehyde 3-phosphate (GAP). In the second phase of glycolysis, only GAP (substrate of the glycolytic enzyme glyceraldehyde 3-phosphate dehydrogenase) continues the pathway, generating two ATP molecules and two NADH molecules. Since two ATP molecules are consumed in the first phase of glycolysis, it can be concluded that up to this moment, there is no net gain of ATPs. However, in the intermediate part of glycolysis, DHAP is isomerized to GAP by the glycolytic enzyme triosephosphate isomerase (TIM or TPI).

Consequently, the isomerization reaction conducted by TIM generates one extra GAP molecule that enters the pathway. This situation results in the production of two additional ATP molecules. Therefore, part of the importance of TIM in glycolysis lies in the extra contribution of glycolytic substrates, which leads to the overall net production of two usable ATP molecules for the cell for each glucose molecule entering the pathway. This is vital for various cancer cells since they highly depend on glycolysis. Under aerobic conditions, this energy pathway can provide up to 60% of the ATP tumor cells demand, whereas, in normal cells, it only represents 10% [74]. Subsequent studies with cancer cells have shown that one or more of the following conditions may exist: impaired mitochondrial function, decreased expression of oxidative enzymes and mitochondrial transporters, as well as truncated tricarboxylic acid cycle, fewer mitochondria per cell, faulty respiratory chain, increased in the number of natural inhibitors of mitochondrial ATP synthase and high sensitivity of mitochondrial DNA to oxidative stress [75].

Undoubtedly, for many types of cancer cells, glycolysis is crucial for generating ATP and is also an essential source of metabolic intermediates for the rapid synthesis of amino acids, nucleotides, and lipids. High levels of aerobic glycolysis have been quantified in a wide range of tumors, regardless of the type of tissue from which they originated. In contrast, normal tissues do not use aerobic glycolysis to meet energy and metabolite demands [63]. This indicates that a characteristic of cancer cells, in combination with unrestricted growth, is to develop an accelerated glycolytic phenotype that provides proliferative advantages to rapidly dividing cells. This is why the glycolytic pathway is still considered an attractive and selective target for developing anticancer compounds. Such a situation has been previously taken advantage of by some researchers, and different works are currently known where the enzymes of this pathway are evaluated as potential targets.

Therefore, developing inhibitors of these enzymes with an anti-cancer effect has focused on using substrate analogs [76,77]. However, this has yet to be successful since it has been shown that they are highly toxic in experimental models as compounds with high affinity towards the active sites of these enzymes are evaluated. These regions are strictly conserved between enzymes, even from different species. Therefore, studying and understanding the biochemical differences (functional and structural) between the glycolytic enzymes of normal and cancer cells are essential for designing and developing new anticancer compounds with more efficient selectivity. Based on the above premises, studying the human glycolytic enzyme triosephosphate isomerase to demonstrate its functional and structural differences under cancer metabolic conditions looks promising. In addition, post-translational changes that commonly occur in this protein seem to be conducive to the development of new strategies for treating a variety of pathologies, including cancer [78].

## 5. Triosephosphate Isomerase Is a Key Metabolic Enzyme

Triosephosphate isomerase is found in almost all organisms and is considered a moonlighting protein (i.e., proteins exhibiting more than one physiologically relevant biochemical or biophysical function). Complete enzyme activity was reported for the dimeric form, even though each monomer has a complete set of catalytic residues. HsTIM is a homodimeric enzyme with five cysteine (Cys) residues per subunit, none directly involved in the catalytic site. This latter is important since cysteines, with their oxidation-sensitive sulfur atom, are responsive to different redox conditions. Thereby, Cys residues have been involved in strategies to target the TIM of several organisms, including in human cells [57,58,59,79].

Among the relevant aspects of the TIM, the interconversion between DHAP and GAP stands out since, as mentioned above, in the final balance of glycolysis, a net gain of two ATP molecules is generated for each glucose molecule processed. Another important aspect of this glycolytic enzyme is that it prevents the accumulation of the DHAP substrate. This molecule can spontaneously degrade to methylglyoxal (MGO), a highly toxic and reactive compound for the cell. DNA, proteins, and lipids are damaged by MGO when it binds covalently to these biomolecules [80]. Absent or inactive TIM has resulted in glucose consumption without ATP gain and DHAP accumulation, leading to high MGO concentrations significantly damaging the cell integrity [81]. Consequently, in cells where glycolysis is essential as a source of energy and metabolites, this condition could seriously threaten the survival of such cells. Moreover, increased DHAP also inhibits hexose-6-phosphate isomerase [82], which in turn highly affects the glycolytic flux in cancer cells [54]. Therefore, based on the above arguments, HsTIM must be considered an attractive target for designing anticancer molecules [56].

## 6. Beyond Being a Target to Impair Energy Metabolism in Cancer Cells, HsTIM Can Be a Methylglyoxal Factory

The impairment of TIM activity does not affect energy metabolism at the system level. Mounting evidence points out that targeting TIM results in the accumulation of DHAP, followed by its chemical conversion into the toxic MGO, leading to the formation of advanced glycation end products (AGEs) [57,78,79,83,84,85,86]. It is known that in cancer cells, aerobic glycolysis occurs 10 to 100 times faster [87]. This produces an increased number of by-products, with 0.1–0.4% of glucose turned into MGO, which is favorable for tumor growth and progression [88]. However, it has already been demonstrated that MGO shows the so-called hormesis process in several types of cancer [89].

Hormesis describes a biphasic dose-dependent response with a low-dose stimulatory or beneficial effect and a high-dose inhibitory or toxic effect [90]. Considering this scenario, MGO would seem to be a double-edged sword in cancer, i.e., it can boost this pathology or become its Achilles heel. Therefore, finding a TIM susceptible to being differentially turned on to overproduce MGO into the pathological cells could be the key to developing specific cancer-cell-directed treatments. Nature has already given us such an opportunity through some PTMs commonly occurring in TIM. Indeed, many features concurring in TIM can facilitate its study to achieve this goal.

### 6.1. Triosephosphate Isomerase Is a Model Molecule for Understanding Structure–Function Relationships

TIM has been widely studied in various species, both functionally and structurally. It is considered a perfect catalyst since it has been estimated that its catalysis is limited only by the diffusion of its substrates [91]. The TIM sequence is known in more than 4900 species (https://www.ncbi.nlm.nih.gov/, accessed on 14 July 2023), as well as the crystallographic structure of more than 90 species of wild-type enzymes and their mutants, alone or in complex with different ligands, inhibitors, or other molecules. The Protein Data Bank (PDB, http://www.rcsb.org/pdb/, accessed on 21 July 2023) database reports more than 905 crystallographic structures related to TIM. The above provides advantages for a more straightforward understanding of this protein in normal and abnormal conditions. Each subunit or monomer of this enzyme is made up of eight central folded β-sheets, surrounded by eight α-helices, which are connected by loops, a folding pattern known as the α/β barrel domain or TIM barrel. The catalytic site is in the central part of the barrel, formed by three amino acids: Lysine (Lys) 14, Histidine (His) 96, and Glutamic Acid (Glu) 166 (Figure 2). This numbering corresponds to the aminoacyl sequence of human TIM (HsTIM, UniProt code P60174-1). Biologically, TIM associates as a dimer, and each monomer has an independent catalytic site; however, the isolated monomer is not active, so dimerization is required to make the enzyme catalytically active. This implies that the interface or contact site between the two subunits is essential for their functionality.

HsTIM expression levels have been correlated with the progression of different types of cancer [92,93]. Further, this enzyme has been pointed out as a potential target in cancer, mainly using inhibitors directed to its active site or factors that diminish its expression [56]. Such a strategy is weak in treating cancer since catalytic centers are the most conserved enzyme regions, and both pathologic and non-pathologic cells show the same enzymes.

HsTIM has been characterized functionally and structurally, and a recent work proposed it as a potential anticancer target [86]. In that study, molecular coupling studies (docking) were carried out, identifying some potential molecules proposed as inhibitors of this enzyme. Despite the excellent contribution of this work, the authors only generated hypotheses that have not been corroborated, and did not define how to obtain inhibitory molecules that have selectivity over the TIM of cancer cells without interfering with the activity of the TIM of normal cells. In seeking out those differences, our group has characterized the deamidation of HsTIM, especially focusing on its relevance in cancer cells concerning normal ones.

### 6.2. Structural Alterations on the Deamidated Triosephosphate Isomerase Are the Base for Rational Drug Design

HsTIM has been extensively characterized for more than 30 years. In particular, a post-translational event known as deamidation has drawn attention. Deamidation is a chemical reaction that irreversibly removes the amide functional group from an organic compound. In proteins, this reaction can occur in vitro or in vivo with individual amide residues depending on the primary sequence, three-dimensional structure, and solution properties, such as pH, temperature, ionic strength, and buffer ions. Deamidation occurs in the aminoacyl residues of asparagine (Asn) and glutamine (Gln). In the case of Asn, the carbonyl carbon of the Asn side chain is attacked by the backbone amide nitrogen atom of the first amino acid residue adjacent to the C-terminal end of Asn, releasing an amide group (deamidation) and forming succinimide. Secondly, hydrolysis of succinimide yields L-aspartic acid (Asp) and L-isoaspartate methyl ester (isoAsp) at an Asp:isoAsp ratio of 1:3. Nonetheless, in some proteins, this ratio can change, resulting in a higher yield of Asp. Deamidation of Gln residues proceeds at a much slower rate than Asn and has been investigated less, though it is likely to follow an analogous chemical trajectory. The mechanism for Gln deamidation yields a glutarimide intermediate, which produces a mix of the L- glutamic acid (L-Glu) and L-isoglutamic acid (L-isoGlu) isomers. Deamidation at neutral pH introduces a negative charge at the deamidation site and sometimes also leads to β isomerization [94,95]. This PTM can progressively alter the proteins’ biological activity and structural integrity. Deamidation is considered in some studies only as a terminal marking for the HsTIM turnover. Although the use of deamidation as a molecular timer has been experimentally demonstrated in some proteins (i.e., cytochrome C and aldolase), this is not the case for HsTIM, where this has yet to be probed in vivo.

Certainly, deamidation is a non-enzymatic event that spontaneously occurs at different speed rates, depending on each protein. Notably, in a few proteins, including TIM, this reaction is accelerated by catalytic cycling [96]. Furthermore, this phenomenon is known to occur naturally in HsTIM. Therefore, the excessive and continuous glycolytic activity in cancer cells may imply a high number of TIM catalytic cycles, potentially leading to an abundance of the deamidated enzyme. Similarly, this would be different from normal cells, as glycolysis is closely controlled in them. In this regard, several studies indicate that cells exhibiting high proliferative activity contain deamidated variants of HsTIM [97]. The amino acid residue Asn 16 is crucial in deamidation and structural rearrangement, particularly in this enzyme. Such a structural rearrangement decreases the thermal stability and increases susceptibility to proteolysis of HsTIM [98]. This residue (Asn 16) aminoacyl is closely related to the catalytic site and the interface in a position of high accessibility to the solvent. However, the deamidation of Asn 72 does not seem to contribute to the general instability of the enzyme. In our previous demonstration [98], we found that imitating deamidation through site-directed mutagenesis causes a notable structural reorganization compared to the non-deamidated enzyme. This supports our prior findings that mimicking such changes can produce comparable results. (Figure 3). Herein, we use the numbering of the primary structure of HsTIM (UniProt code P60174-1).

Figure 3 shows the structural overlap of wild-type HsTIM (WT) versus deamidated HsTIM (N16D). This analysis demonstrates that one of the dimer subunits in the deamidated enzyme, when overlapping with the non-deamidated enzyme, is displaced by an average of 5.8 Å. Reviewing the crystallographic structure of the deamidated HsTIM, it was identified that approximately 60% of its interactions are lost at the dimer interface, thus generating a large porosity in that area. Consequently, and due to the structural affectation, the deamidated HsTIM greatly increases its accessibility to small molecules, unlike the non-deamidated enzyme, which is not accessible to these molecules. This is demonstrated both by computational methods such as molecular docking (Figure 4) and experimentally.

Some time ago, we showed that the reactive thiol drug omeprazole could inhibit the recombinant mutated enzyme HsTIM N16D (which imitates the previously deamidated HsTIM) [78]. In contrast, the recombinant enzyme HsTIM WT (non-deamidated) retained its activity without being affected by the mentioned drug. A bacterial system lacking endogenous TIM was used to determine the usefulness as a target of the deamidated HsTIM under a cellular environment to obtain selective cytotoxicity. Bacteria were transformed with the gene encoding for non-deamidated HsTIM (WT) or deamidated HsTIM (N16D). This bacterial model, complemented by non-deamidated or deamidated HsTIM, demonstrated that the exposition to omeprazole inhibits cell growth in bacteria carrying the *hstim n16d* gene (producing deamidated HsTIM). In contrast, bacteria transformed with the *hstim wt* gene (producing non-deamidated HsTIM) were not disturbed in their cell growth, even in the presence of omeprazole [78]. These results indicate notable variations in response to the drug in bacteria that produce either WT or N16D.

On the other hand, deamidated HsTIM is expected to be hardly detectable in normal human cells, most likely due to protein turnover. Consequently, proving that this deamidated enzyme could exist and accumulate in human diseases where cells relied heavily on glycolysis was important. For this purpose, we used MDA-MB-231 breast cancer cells since they are highly glycolytic. From our experiments, we discovered that rabeprazole and auranofin, which are thiol-reactive drugs, effectively prevent breast cancer cell proliferation. Moreover, we observed that these drugs also inhibit the activity of the cellular HsTIM enzyme. The drugs did not affect cell proliferation or HsTIM enzymatic activity when exposed to normal HMEC cells (of mammary epithelium) [79]. Subsequent experiments helped to identify the deamidated enzyme accumulation in cancer cells and its absence in normal cells. (Figure 5 and Figure 6). Hydroxylamine protein hydrolysis is an easy and cost-effective method to identify the presence or absence of deamidation in protein samples, generating various peptide fragments that can be identified by their hydrolysis profile on SDS-PAGE denaturing gels. Hydroxylamine can selectively cleave the intermediary succinimide formed by the side chains of Asn and Gly in non-deamidated proteins under alkaline conditions. However, in deamidated proteins, the Asp-Gly (or isoAsp-Gly) chain does not form succinimide under the same conditions. This shows that the deamidated peptide cannot be cleaved.

Since the sequence of HsTIM contains two Asn-Gly pairs at positions 16–17 and 72–73, cleavage might generate three peptides when the enzyme is not deamidated. When the cleavage of non-deamidated HsTIM is incomplete, a peptide with a size of 7.83 kDa could also be produced. The resulting peptides expected for a non-deamidated HsTIM are 26.9 (the non-cleaved monomer), 1.87, 5.96, 18.87, and 7.83 kDa (see Figure 5, lanes 2 and 5). On the other hand, the cleavage of once deamidated HsTIM only produces peptides with sizes of 7.83 and 18.87 kDa (Figure 5, lane 3), whereas twice deamidated HsTIM would not be cleaved (Figure 5, lanes 4 and 6).

Different degrees of hydrolysis of the recombinant non-deamidated enzymes HsTIM (WT), once deamidated HsTIM (N16D), and twice deamidated HsTIM (N16D/N72D) are shown in Figure 5. Recombinant enzymes serve as a reference to verify the presence or absence of deamidated HsTIM in normal and cancer cells. Lanes 5 and 6 show the hydrolysis profile of HsTIM extracted from normal cells (HMEC) and cancer cells (MDA-MB-231), respectively. The HsTIM from normal cells (lane 5) presents a hydrolysis profile like the non-deamidated HsTIM (lane 2). In contrast, the HsTIM from cancer cells (lane 6) produces a hydrolysis profile like that of the deamidated HsTIM (lane 3). Therefore, this method effectively shows the presence of the deamidated enzyme in cancer cells. To reinforce these findings, native (non-denaturing) gels are helpful in the search for the presence of negatively charged isoforms, as occurs when HsTIM is deamidated. In this assay, it was also necessary to use the recombinant enzymes as a migration reference and total protein extracts from cancerous and normal cells in the absence and presence of rabeprazole. By revealing the HsTIM signal by Western blot, we demonstrated the presence of acidic isoforms in breast cancer cells, while in normal cells, such isoforms were not observed (Figure 7).

Figure 6 shows both the presence of acidic isoforms (with negative charges) and the accumulation of deamidated HsTIM in breast cancer cells. By contrast, such acidic isoforms were not observed in normal cells. Additionally, treatment with rabeprazole increased the accumulation of acidic isoforms in the cancer cell line (Figure 6, lanes 5 and 4, respectively, in cancer cells). In summary, the selective inactivation of HsTIM in cancer cells, with the corresponding cell death in the presence of the reactive thiol drug, and the presence and accumulation of deamidated HsTIM, mainly in breast cancer cells, allowed us to identify a molecular target that is present in cancer cells but do not in normal cells. Based on the cell-selective reaction observed with thiol-reactive drugs, we consider this a significant step in searching for anticancer alternatives with high selectivity, safety, and low costs. Other authors have recently achieved similar results but through more advanced and costly technologies than the ones we employed [99]. Fortunately, this serves to strengthen our previously established conclusions.

## 7. A Group of PTMs in HsTIM Could Be Considered a Target for Cancer Therapies

There are a variety of PTMs for proteins, with 461 different types of modified amino acids registered for eukaryotic proteins in UniProt. PTMs impact protein activity, stability, location, and interactions [100]. Ten different types of PTMs out of 29 HsTIM amino acid residues have been reported or predicted so far (Table 1). Some of these modifications can affect its structure and function and could represent the bases to develop compounds with anticancer potential. Unfortunately, knowledge of the characteristics of these PTMs is scarce for most of them. Nonetheless, after analyzing the studies reported on this subject for HsTIM, at least three of these PTMs seem promising to us for consideration as molecular targets in cancer.

### 7.1. Phosphorylation in HsTIM

An important PTM that has been identified in HsTIM is phosphorylation, which involves the addition of a phosphate group to amino acid residues such as serine (Ser), threonine (Thr), or tyrosine (Tyr). Kinase enzymes perform this process and can occur in response to various cellular signals. Consequently, phosphorylation can cause conformational changes in HsTIM, altering its topology and creating new binding sites for small molecules. For example, HsTIM phosphorylation has been observed in response to various stimuli, such as stress, growth factors, etc. In lung cancer, phosphorylation of Ser 58 by kinase PRKACA has been identified as essential to maintain cancer growth and metastasis [104].

### 7.2. Heterodimers of HsTIM Phosphorylated at Ser21 Are Present in Cancer Cells

Various publications show that HsTIM is phosphorylated at multiple sites, causing either a decrease in its enzymatic activity, as demonstrated in HeLa cancer cells [119], or an increase in its enzymatic activity, as occurs in lung cancer and recombinant S21E proteoform [104,120]. Furthermore, it has been shown that phosphorylation can modulate HsTIM enzyme activity and even its sub-cellular distribution. Duan et al. in 2020 found that HsTIM is phosphorylated at Serine 21 (although they referred to it as Serine 58 since they were based on the HsTIM isoform 2) and identified that such phosphorylation significantly increased both production and activity of this enzyme in some types of tumors. They also showed that blocking HsTIM phosphorylation partially inhibits glycolysis, cancer cell growth, and metastasis. Therefore, it is highly probable that HsTIM phosphorylation could generate structural changes that make differences with the non-phosphorylated enzyme, as occurs in the case of deamidated HsTIM. As the in silico modeling predicts, this type of phosphorylation can cause structural alterations in the enzyme (Figure 7). After phosphorylation, contacts not allowed in this area are generated, potentially leading to substantial structural alterations making the enzyme core accessible to small molecules that could interfere with its function.

In the case of Serine 21 phosphorylation, which was found in cancer cell lines (referred to as Ser 20 by the authors due to the lack of Met 1), it has already been experimentally shown that HsTIM increases its enzymatic activity by forming heterodimer complexes, which helps glycolysis to generate more ATP [120]. Due to the above and in the context of designing molecules with anticancer potential, the altered structure of phosphorylated HsTIM could be exploited as a pharmacological target. Cancer cells may present abnormal signaling pathways involving aberrant phosphorylation of specific proteins such as HsTIM. Thus, by designing compounds that selectively target these altered sites, it would be possible to inhibit HsTIM activity, which can prevent cell proliferation and thereby prevent the growth and propagation of cancer cells. Furthermore, this enzyme might not be phosphorylated (or to a lower degree) in normal cells. So, this phenomenon presents an opportunity to develop compounds that specifically target this type of protein without affecting healthy cells. The incoming challenge is demonstrating that such phosphorylation occurs only or preferentially in cancer cells rather than normal cells.

As is known, continuous catalytic cycles promote the deamidation of HsTIM [97,121]. Therefore, in line with our findings on the deamidation of HsTIM, one can expect that the phosphorylated and highly active HsTIM described in cancer cell lines [120] would be prone to be deamidated. This can turn cells with phosphorylated HsTIM into cell-specific targets in cancer due to their propensity to be deamidated, as previously demonstrated [79].

### 7.3. S-nitrosylation in HsTIM

S-nitrosylation is a selective covalent PTM that adds a nitrosyl group to the reactive thiol group of Cys to form S-nitrosothiol, which is a key mechanism in transferring nitric oxide-mediated signals. Multiple human diseases, including cancer, show abnormal nitric oxide levels. S-nitrosylation is also a relevant PTM identified and studied in HsTIM and TIMs from other organisms [116]. It has been shown that this modification can modulate the activity and stability of this enzyme. For example, some studies have shown that S-nitrosylation can regulate HsTIM activity, which has been implicated in cancer development and progression. Further, it was demonstrated that S-nitrosylation decreases HsTIM enzymatic activity [116]. Additionally, the crystallographic structure of this enzyme was determined, where it was possible to identify the Cys 217 S-nitrosylated [116](Figure 8).

The evident structural alteration promoted by this post-translational modification gives us a guideline to consider this enzyme in developing compounds with anticancer potential. Additionally, in an altered environment such as that of cancer cells, this type of enzyme could accumulate, as has been demonstrated for the S-nitrosylation of HsTIM, where its stability was affected since it is more resistant to degradation [122]. This increased stability could contribute to an accumulation of HsTIM in cancer cells, promoting cancer cell survival and proliferation. There are many studies where this glycolytic enzyme is S-nitrosylated. For example, it was found in human pancreatic ductal adenocarcinoma [123]. Additionally, it was proposed for several proteins in metastatic cancer cells that S-nitrosylation can induce conformational changes, influencing protein stability and protein-protein interactions [124,125]. Such structural changes might also be occurring in S-nitrosylated HsTIM. Therefore, as demonstrated for deamidated HsTIM, those changes can yield conformational rearrangements that make HsTIM more accessible to small molecules.

### 7.4. S-glutathionylation in HsTIM

S-glutathionylation refers to the specific post-translational modification of protein cysteine residues by the reversible covalent addition of glutathione. This modification generally rises during oxidative/nitrosative stress but might also rise in unstressed cells under normal physiological conditions [126]. It is reported that S-glutathionylation of Cys217 leads to the inhibition of *Plasmodium falciparum* TIM [127]. Since Cys217 is a conserved residue in mammalian TIM, it may be considered an interesting target to regulate TIM activity by glutathionylation in cancer [56]. On this line, it was found that the TIM of human T lymphocytes undergoes glutathionylation under oxidative stress [128].

Finally, as we have seen, these post-translational events in HsTIM, such as deamidation, phosphorylation, S-nitrosylation, and S-glutathionylation can alter the structure and function of this enzyme, which is critical in the glycolytic pathway. Therefore, these subtle changes in the structure of HsTIM resulting from these modifications can represent the basis for developing new molecules with anticancer potential. Concerning the above, and the fact that this enzyme is up-regulated in many cancer cells, there will be great possibilities for finding differences concerning normal cells. However, this is not limited to HsTIM. Many proteins with structural alteration due to post-translational modifications will emerge as an army of new opportunities for better tools in the fight against cancer. Those proteins that meet the requirements presented for the HsTIM could be considered for this purpose.

While we were writing this review, two papers were published supporting our hypothesis that HsTIM undergoes these PTMs in cancer cells. These works have found that HsTIM, with these modifications, accumulates in certain types of cancer and can play an essential role in the mechanistic explanation of cancer development [99,129].

## 8. Conclusions

Even though the study of cancer is one of the most prolific fields in all of science, it is undeniable that eradicating it remains a significant challenge. Scientists and physicians alike are working tirelessly towards finding a cure. On this line, it is widely recognized that therapies targeting biomolecules related to the glycolytic pathway are essential in the fight against cancer, as this pathway is considered a hallmark of this group of diseases. However, the biggest weakness of such strategies is that they are mainly based on the abundance differences of specific molecules between normal and cancer cells. Additionally, using substrate analogs as cancer medication is likely unsuccessful because the catalytic sites are highly conserved across enzymes.

As if from the dark side of the moon, PTMs are emerging to be nominated as the best supporting actor (maybe the best actor?) in maintaining the malignant processes of cancer cells. Neglected until now, PTMs commonly occurring in glycolytic proteins are shyly showing their potential to be used as targets to cope with cancer, possibly more efficiently and safely than current therapies. In this regard, human triosephosphate isomerase can be awarded because it owns multiple amino acid positions susceptible to being post-translationally modified. Some of those PTMs in HsTIM can promote structural rearrangements in this protein that might act as double-edged swords. This means that, on the one hand, these structural changes underlie the production of the oncometabolite MGO. Cumulative research has focused on the emerging role of MGO as cellular oncometabolite involved in tumorigenesis-associated proliferative control, redox dysregulation, and epigenetic recoding [130,131]. Therefore, PTMs, HsTIM structural rearrangements, and MGO production represent a tri-partite component essential in maintaining the malignancy of cancer. It should not be forgotten that an unusual increase in post-translationally modified HsTIM is required to maintain the status quo of the cancer cell. Therefore, to maintain the cancer cell’s dominance over other cells in the body, HsTIM relies on an intricate system of molecular components to aid in its complex mission. Further studies will provide more clarity on this intricate metabolic system. Metabolomics and the development of highly sensitive and efficient mass spectrometry and proteomics will expand our understanding of this problem. This could be further leveraged to improve the management of patients with cancer.

On the other hand (the light side of the moon), the structural rearrangements caused by PTMs on HsTIM also turn this protein accessible to small molecules to which it previously was unavailable as a target, as was already demonstrated for deamidated HsTIM in vitro and in vivo in mammary cancer cells [79]. This research showed that the small molecules assayed against deamidated HsTIM provoked an exacerbated production of MGO, leading to cancer cell death. Here, we have shown elements to consider that, in theory, phosphorylation, nitrosylation, and glutathionylation could exert similar effects in HsTIM to those caused by deamidation. We look forward to studies that unravel the potential of these PTMs to suggest new strategies to treat cancer.

## Figures and Tables

**Figure 1 molecules-28-06163-f001:**
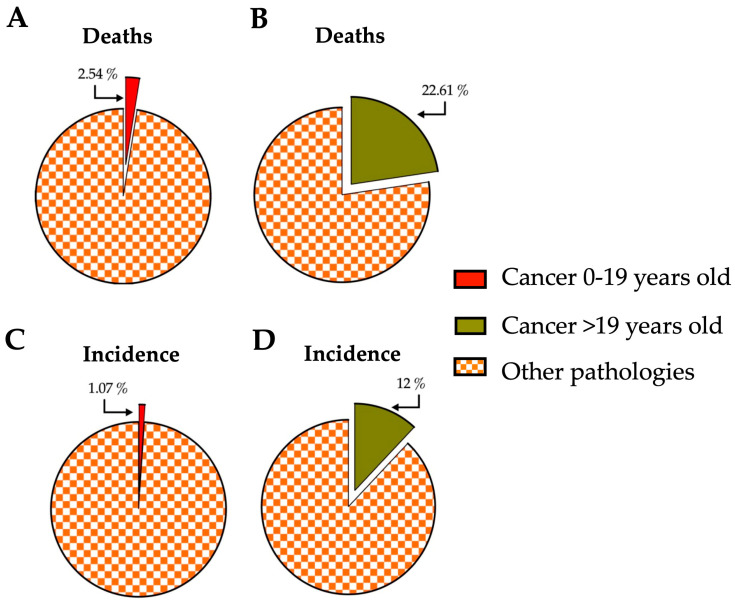
Impact of cancer on young and older people concerning deaths and incidence compared with other pathologies affecting humanity globally. (**A**,**B**) The percentage of deaths related to those caused by cancer compared with the rest of the pathologies reported for the young (0–19 years old) and older (>19 years old) population. (**C**,**D**) The percentages represented by the incidence of cancer compared to the sum of incidences of all pathologies reported for the young (0–19 years old) and older (>19 years old) population. The data correspond to the year 2019 and were obtained from https://vizhub.healthdata.org, accessed on 14 July 2023. The number of deaths and incidences corresponding to violence or accidents were excluded.

**Figure 2 molecules-28-06163-f002:**
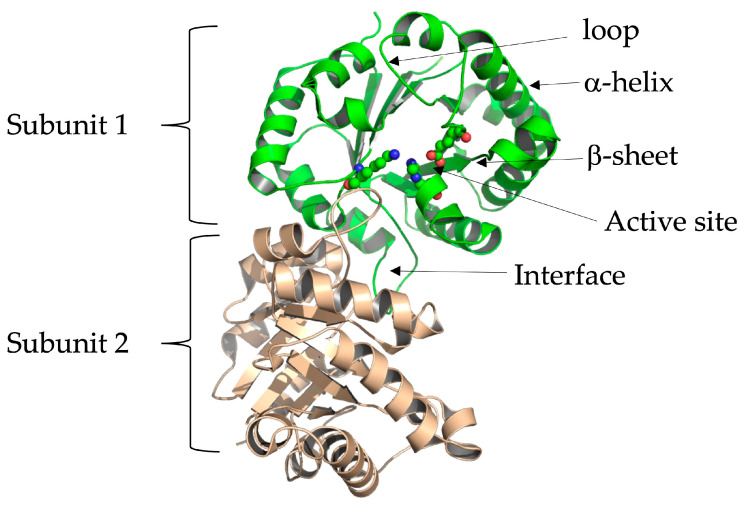
3D structure in loops and ribbons format of human triosephosphate isomerase’s (HsTIM) crystallographic coordinates, PDB code 2jk2. The biological unit of the enzyme is a dimer (green and brown). The interface (contact site between two subunits) and the position of the amino acids constituting the active site in one of its subunits are shown. We created and modeled the figure with PyMOL, version 2.0.7 (Schrödinger, LLC, New York, NY, USA).

**Figure 3 molecules-28-06163-f003:**
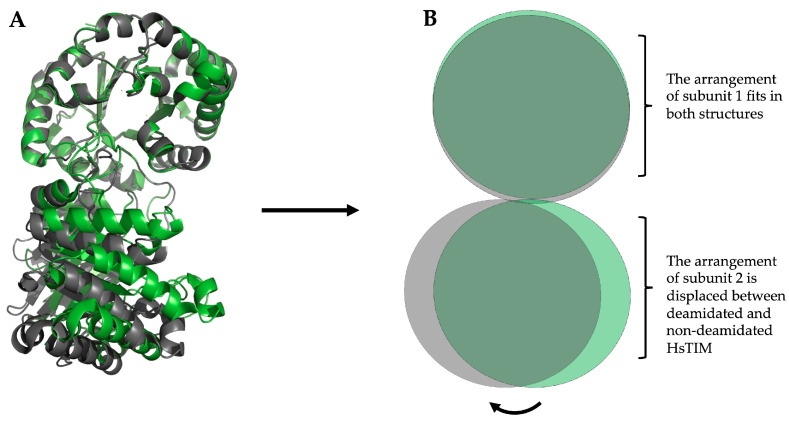
Structural overlapping of the HsTIM dimer, non-deamidated (WT), and deamidated (N16D). (**A**) Representation of the crystallographic structure in loops and ribbons. Overlay of the non-deamidated HsTIM, PDB code 2jk2 (green) versus deamidated HsTIM, PDB code 4unk (grey). In both structures, one subunit aligns correctly, while the other is displaced by an average of 5.8 Å, concerning the non-deamidated HsTIM. (**B**) Schematic representation of the overlapping is shown in (**A**). This scheme highlights the structural rearrangement that causes the displacement of one of the HsTIM subunits due to deamidation. We created and modeled the figure with PyMOL, version 2.0.7 (Schrödinger, LLC, New York, NY, USA).

**Figure 4 molecules-28-06163-f004:**
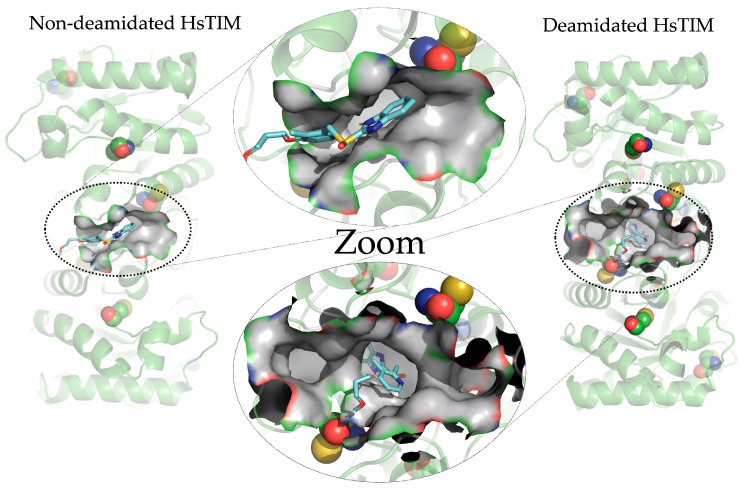
Crystallographic structures of the non-deamidated HsTIM [PDB code: 2jk2] and deamidated HsTIM [PDB code: 4unk] in loops and ribbons. The main cavity at the interface and how the rabeprazole molecule is coupled are shown. The detail of the molecular coupling in both structures is shown in the center of the image. We created and modeled the figure with PyMOL, version 2.0.7 (Schrödinger, LLC, New York, NY, USA).

**Figure 5 molecules-28-06163-f005:**
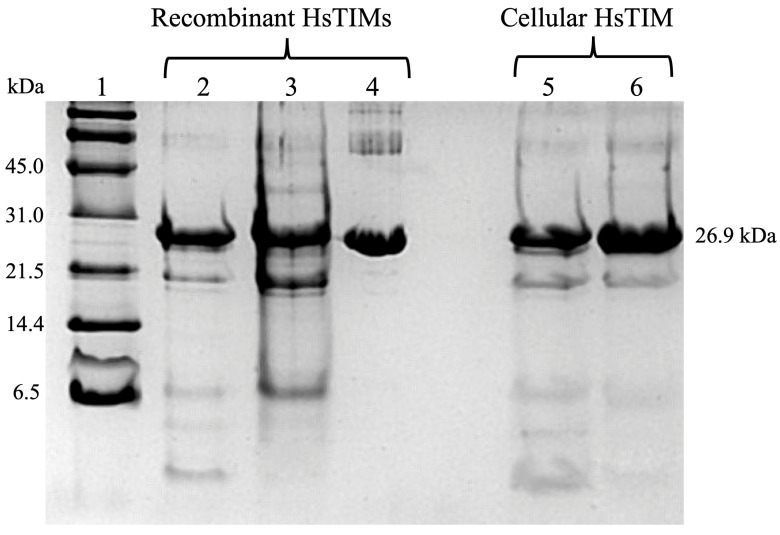
Representative study to identify deamidated HsTIM in breast cancer cells. SDS-PAGE at 16% hydroxylamine hydrolysis of recombinant HsTIMs and cell-derived TIMs: Lane 1: molecular weight standard, lanes 2–4: (10 µg protein/lane) non-deamidated (WT) HsTIMs, once deamidated (N16D) and twice deamidated (N16D/N72D), lanes 5 and 6: (20 µg protein/lane) HsTIM immunoprecipitated from normal and cancer cells, respectively. The figure is unpublished and was performed with standardized methods from our laboratory (Enriquez-Flores et al. [79]).

**Figure 6 molecules-28-06163-f006:**
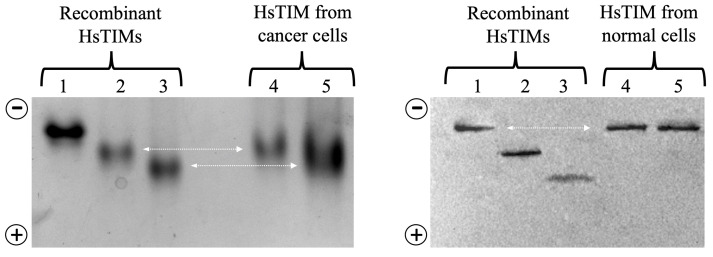
Representative study by western blot of recombinant and cellular HsTIM in native gels. Lanes 1 to 3 (2 µg protein/lane), recombinant HsTIM not deamidated, once deamidated and twice deamidated, respectively. Lanes 4 and 5 (100 µg protein extract/lane), protein extract from cancer cells (left gel) and normal cells (right gel). Lanes 4: controls, lanes 5 treatment with rabeprazole. The arrow indicates the level at that TIM migrated from cancer and normal cells relative to the reference (recombinant) proteins. The gels’ polarity is indicated on each figure’s left side. The figure is unpublished and was performed with standardized methods from our laboratory (Enriquez-Flores et al. [79]).

**Figure 7 molecules-28-06163-f007:**
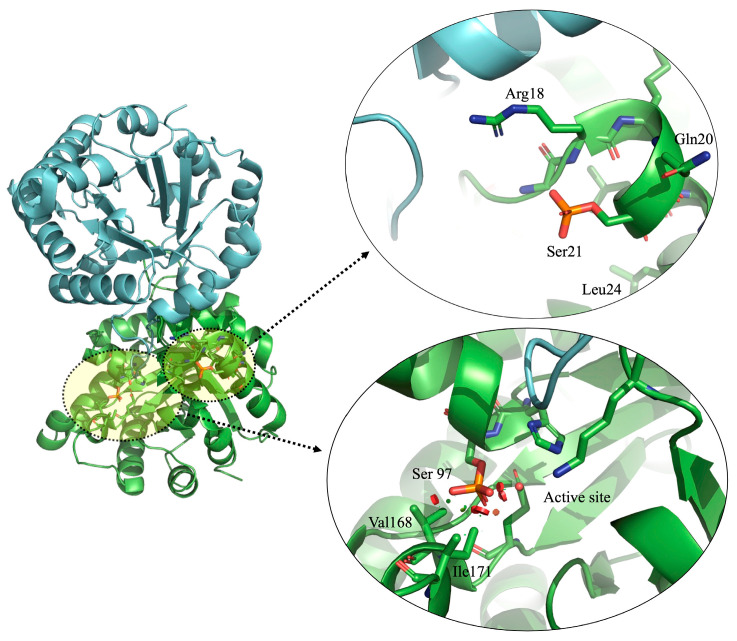
Crystallographic structure of the HsTIM [PDB code: 2jk2] in loops and ribbons. The two subunits of the dimer are shown in cyan and green colors. In one of the monomers, phosphorylated Serine 21 and 97 are observed (overlaid in light yellow). The right side of the figure is an enlarged view of the phosphorylated amino acids. In silico phosphorylation and structure were created and modeled by us with PyMOL, version 2.0.7 (Schrödinger, LLC, New York, NY, USA).

**Figure 8 molecules-28-06163-f008:**
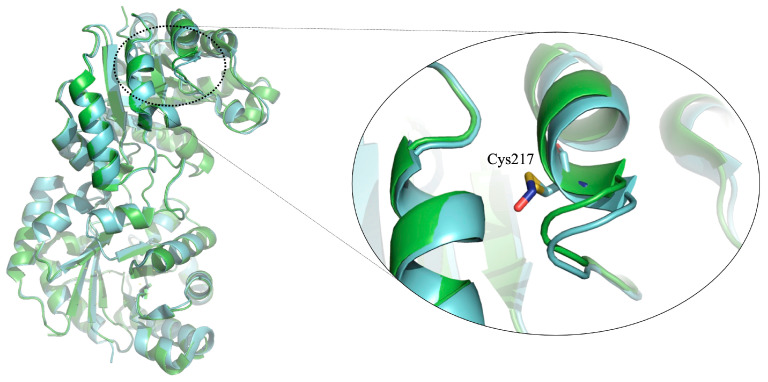
Superimposed crystallographic structures of HsTIM WT [PDB code: 2jk2] and HsTIM S-nitrosylated [PDB code: 6d43] in loops and ribbons. The two subunits of the dimer are shown in cyan and green colors. The structural overlapping of the dimer of the mentioned enzymes is shown (WT green and S-nitrosylated cyan). Zoom of the 217 S-nitrosylated cysteine area is shown on the right side. A slight structural alteration promoted by post-translational modification is observed. We created and modeled the figure with PyMOL, version 2.0.7 (Schrödinger, LLC, New York, NY, USA).

**Table 1 molecules-28-06163-t001:** Post-translational modifications reported in triosephosphate isomerase.

Positions of Modified Amino Acid Residues	Post-Translational Modification	Reference
14	N6-acetyllysine	[101]
16	Deamidation	[79]
21	Phosphoserine	[102]
28	Phosphothreonine	[103]
58	Phosphoserine	[104]
59	Succinylation	[105]
68	3′-nitrotyrosine	[103]
69	Ubiquitylation	[106]
71, 76	Phosphothreonine	[103]
80	Phosphoserine	[107]
90	Phosphothreonine	[103]
97	Phosphoserine	[103]
106	Phosphoserine	[108]
135	Methylation	[109]
149, 156 *	N6-acetyllysine	[110]
159 *	Phosphoserine	[111]
165 **	3′-nitrotyrosine	[112]
173	Phosphothreonine	[113]
194	N6-acetyllysine	[105]
196	N-glycosylation	[114]
204	Phosphoserine	[108]
209 **	3′-nitrotyrosine	[112]
212	Phosphoserine	[108]
214	Phosphothreonine	[115]
217	S-nitrosylation	[116]
217	Glutathionylation	[117]
225	N6-acetyllysine	[118]
238	N6-acetyllysine	[105]

Note: The numbering of amino acid positions corresponds to that of the primary structure. * Reported in homologous triosephosphate isomerase. ** In silico predicted.

## Data Availability

Not applicable.

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
