# Peer review of "Human Triosephosphate Isomerase Is a Potential Target in Cancer Due to Commonly Occurring Post-Translational Modifications"

_molecules, 2023, doi:10.3390/molecules28166163_

Round 1

Reviewer 1 Report

The topic of this review manuscript is interesting, and the manuscript is timely, especially because there are very few review antecedents for a similar topic. The following search in PubMed

((triose phosphate isomerase[MeSH Terms]) AND (cancer[MeSH Major Topic])) AND (review[Publication Type])

implemented without any time limit, gave only two results. One in 2020 entitled “Therapeutic Targeting of Cancer Metabolism with Triosephosphate Isomerase” by Pekel and Ari (PMID: 32180338, cited in the manuscript as reference 41). Actually, these authors state that “Here, we reviewed the relationship between TPI and cancer for the first time”. The other result is a 2008 review entitled “Glycolytic enzyme inhibitors in cancer treatment”, i.e. less specifically related to triose phosphate isomerase.

The specific interest of the current review relies in that it is centered on the posttranslational modifications of the enzyme and the opportunity they offer to develop drug treatments targeted to triose phosphate isomerase.

Below, I am pointing to aspects of the manuscript that need to be improved.

MAJOR POINTS

1.- The timely character of the manuscript, according to the PubMed search above described, should be highlighted in the Introduction after the modification described in point 2.

2.- The introduction of the review (lines 30-105), with Figures 1 and 2, and the section on diagnosis (lines 109-124), are too general, not directly related to the question reviewed in the manuscript. They should be deleted and changed to a more direct presentation of triose phosphate isomerase. The sections on cancer treatment and cancer metabolism help to this end.

3.- The sources for manuscript Figures 3–9, including molecular models, should be always clearly stated. This is not always evident in the text or in the legends. Figure 6 is most likely originating from reference 44, but it is not exactly the same as Figure 2a in ref. 44. Figure 7 seems similar to a part of Figure 2b in ref. 44, but not exactly. This should be cleared out to avoid the impression that the review contains unpublished experimental results.

4.- In lines 416-417, it is mentioned that twenty posttranslational modifications of 6 different types on 17 residues of HsTIM have been reported until today [59]. This is interesting information which is dificult to recover from the literature. A new Table with a list of these modification would be helpful to the readers.

5.- In line 319, N16D substitution is mentioned as coming from reference 21. However, in this reference, the substitution is N15D. This is either a mistake or needs explanation.

6.- The hydroxylamine protein hydrolysis method used to identify deamidated enzyme should be explained, as Figure 6 and comments on lines 368-372 and 380-390 are insufficient.

8.- A Conclusions section would be welcome.

Author Response

Reviewer 1

Comments and Suggestions for Authors

 The topic of this review manuscript is interesting, and the manuscript is timely, especially because there are very few review antecedents for a similar topic. The following search in PubMed

 ((triose phosphate isomerase[MeSH Terms]) AND (cancer[MeSH Major Topic])) AND (review[Publication Type])

 implemented without any time limit, gave only two results. One in 2020 entitled “Therapeutic Targeting of Cancer Metabolism with Triosephosphate Isomerase” by Pekel and Ari (PMID: 32180338, cited in the manuscript as reference 41). Actually, these authors state that “Here, we reviewed the relationship between TPI and cancer for the first time”. The other result is a 2008 review entitled “Glycolytic enzyme inhibitors in cancer treatment”, i.e. less specifically related to triose phosphate isomerase.

 The specific interest of the current review relies in that it is centered on the posttranslational modifications of the enzyme and the opportunity they offer to develop drug treatments targeted to triose phosphate isomerase.

 Below, I am pointing to aspects of the manuscript that need to be improved.

Response: Thank you for taking the time to review our manuscript. We value your feedback and have made the necessary changes. You can find the updated version of the manuscript with all the changes highlighted in red. We hope you find the updated version to your liking.

MAJOR POINTS

1.- The timely character of the manuscript, according to the PubMed search above described, should be highlighted in the Introduction after the modification described in point 2.

2.- The introduction of the review (lines 30-105), with Figures 1 and 2, and the section on diagnosis (lines 109-124), are too general, not directly related to the question reviewed in the manuscript. They should be deleted and changed to a more direct presentation of triose phosphate isomerase. The sections on cancer treatment and cancer metabolism help to this end.

 Answer: Thank you for your comment. We rewrite the introduction considering your advice and that from reviewer 2, thereby increasing references. The previous figures 1 and 2 were deleted, and a new figure 1 was added. The numbering of figures was adapted to that.

3.- The sources for manuscript Figures 3–9, including molecular models, should be always clearly stated. This is not always evident in the text or in the legends. Figure 6 is most likely originating from reference 44, but it is not exactly the same as Figure 2a in ref. 44. Figure 7 seems similar to a part of Figure 2b in ref. 44, but not exactly. This should be cleared out to avoid the impression that the review contains unpublished experimental results.

Answer: We very much appreciate this observation. Although we already published our results about deamidated HsTIM (previously reference 44, now reference 79 in the corrected version), the figures included in this manuscript are unpublished. The figures contain the results of recent assays with methods standardized in our laboratory. Models were performed based on the information gathered from the literature but incorporated into an in silico representation made in our laboratory. Based on your request, we stated this in the legend of each figure.

4.- In lines 416-417, it is mentioned that twenty posttranslational modifications of 6 different types on 17 residues of HsTIM have been reported until today [59]. This is interesting information which is dificult to recover from the literature. A new Table with a list of these modification would be helpful to the readers.

 Answer: Based on this request, we deeper search for post-translational modifications reported or predicted for HsTIM. The new Table 1 shows the results of this search.

5.- In line 319, N16D substitution is mentioned as coming from reference 21. However, in this reference, the substitution is N15D. This is either a mistake or needs explanation.

 Answer: Some works use the numbering from the crystallographic coordinates of HsTIM. That numbering does not consider the methionine at position number one. We consider that the correct numbering is that of the primary structure (UniProt code P60174-1). That is why we consistently refer to position 16 instead of 15. A note to clarify this was added.

6.- The hydroxylamine protein hydrolysis method used to identify deamidated enzyme should be explained, as Figure 6 and comments on lines 368-372 and 380-390 are insufficient.

Answer: Done.

8.- A Conclusions section would be welcome.

Answer: Done.

Reviewer 2 Report

The article entitled « The human triosephosphate isomerase is a potential target in cancer due to commonly occurring posttranslational modifications » is authored by Sergio Enríquez-Flores, Ignacio De la Mora-De la Mora, Itzhel García-Torres, Luis A. Flores-López, Yoalli Martínez-Pérez and Gabriel López-Velázquez.

This article is a review which proposes an overview of the the possibility to use human triosephosphate isomerase, and in particular its post-translational modifications, as an intervention point in cancer therapy.

The manuscript starts with general information on cancer and its burden on health in modern societies. It then briefly speaks on the techniques available and commonly used for diagnosis, which is followed by another short paragraph on the current options employed to treat patients. The article then dives into the metabolic modifications observed in most cancer cells, and explains how the glycolytic pathway could be used as a weakness to target cancer cells while sparing normal cells. The final part of the manuscript focuses on triosephosphate isomerase, its very particular function in glucose metabolism in cancer cells, and the rationale to turn it into a pharmaceutical target to fight cancer. It then reviews the current knowledge on the enzyme structure, how post-translational modifications alter the structure/function of the enzyme, and how this differs in cancer cells compared to normal cells, thus making the point of using those in cancer therapy.

The review is rather well structured with a classical progression from general data toward very specific points but the organization of the manuscript could be improved as will be proposed below. Some wordings are occasionally weird and absolutely require editing by a native English speaker. The first part of the manuscript seems to have been written by someone less expert than the author of the second part, but homogenizing the work should not take too much effort. The part on metabolism is really well explained, along with the part on HsTPI, which collectively make this review a valuable work.

Detailed analysis:

The general description of cancer does not provide any original angle and some of the points raised even leave the reader dubious. For example, the claim that “cancer has often been conceptualized wrongly as a disease of old age” (line 54). To make their point, the authors explain that cancer is a significant cause of death in children aged from 1 to 14 (line 57), and that there are more and more new cases of cancers detected in children (line 59). My first objection to this is that it was never said that cancer is “exclusively” a disease of old age. It is instead common knowledge that cancer strikes every age. My second objection is that if the authors are determined to make a point on this, the data they should provide are what part cancer plays in all pathologies in old age, and what part cancer plays in all pathologies in young age, rather than providing data to say that cancer exists in children as well.

Line 68: the authors explain that advancing knowledge on cancer at the molecular scale “has not yet significantly impacted the reduction of new cases”. It is my understanding that curing cancer is far more realistic than aiming to prevent it. So in that respect, the number patients treated every day show that scientific advances deliver what they are expected to, even if better treatments but be found.

I would suggest to combine figure 1 and 2 for a comparison at a glance.

In the paragraph explaining the causes of cancer, I would suggest to use the established term “epigenetic” to encompass the examples provided line 92-93.

The sentence line 97-98 eludes me. How could risk factors limit cancer development? The sentence should probably be written the other way around.

The paragraph dedicated to cancer diagnosis should be improved. It is too brief and does not provide any meaningful addition to the manuscript in its current state. It should at least state that cancer diagnosis relies on biomarkers (may be list a few), and that they may be investigated at the level of DNA or at the level of proteins, with the techniques indicated.

Same thing on the paragraph speaking about cancer treatment. It is not very meaningful and should be expended. I understand that its purpose is to draw attention to the metabolic differences between cancer cells and normal cells. This sentence should be the last of this paragraph to make the transitions to the next one.

Same thing again on the paragraph on cancer cell metabolism. With the state of the art on this topic, the paragraph could definitely be improved. In its current version, it is too vague. One angle could be a historical perspective? Another angle could be the different types of metabolic changes observed depending on cancer types? But it should definitely be made more specific.

Paragraph 5 is where the review really starts for me. Lines 194-195 do not say much about the potential complexity of metabolic rewiring during cancer. I understand that it is not the purpose of this review, but at least the authors should point to one of the many reviews in the field at the end of this sentence.

Same remarks for line 205-206. Provide some examples with accompanying references.

The paragraph leads to HsTIM, but its deamidation is only mentioned very briefly and I think it might be confusing for lay people to be introduced to a concept that will only be developed in paragraph 7.2.

In paragraph 7.2, the description of deamidation is not correct as some products of the reaction are missing. Deamidation of Asn produces a mixture of Asp and IsoAsp residues, while that of Gln produces a mixture of Glu and IsoGlu. Please explain the meaning of line 302: do the authors mean that deamidation-induced increased turnover of proteins is controversial, or that the outcome of deamidation on protein stability depends on the protein studied? The latter is certainly true in cases were deamidation leads to gain of functions like that of ceruloplasmin in neurodegeneration.

Line 306: please modify to “depending on each protein”.

Line 320: what is a structural involvement? This is not an established term.

Line 336-337 and 442: same remark with the term “HsTIM is highly permeable”. Permeability describes the ability to cross a barrier. May be a better suited word would be accessibility of TIM to small molecules?

I am very surprised to find experimental results in a review. I have never seen this before and the format of a review does not transpose to research articles because there is not material and method section. This point is actually what disturbs me most in this whole work. A review article does not contain pieces of research (which would, by the way, be very valuable material for a true research article!).

Finally I would recommend to reorganize paragraph 8 and to break it down into one section for each post-translational modification. Also, there should be a paragraph “conclusion” summarizing the manuscript and pointing to future directions.

Minor points:

- modify posttranslational to most-translational in the title and line 20.

- unclear about the “necessary” line 21.

- line 31: originating FROM

- line 32 : weird wording “which can affect humans from mild to very serious”

- line 33: unclear about “after a period”. What does it mean?

- line 33: weird wording “divide, and duplicate their number” is redundant.

- line 43: weird wording “ the WHO has pointed out the preceding”?

- line 167: energy-demanding

- line 168: unclear about the “used to”?

- line 226: “both” means that there are 2 subjects. Here there are three: DNA, proteins and lipids.

- line 227: the sentence “covalently…” has no verb. Same in line 300.

- lines 274-278: should be formatted as legend.

- line 350: weird wording “whether a cell containing deamidated… could be selectively disturbed”.

- line 451: referred to AS.

Some wordings are occasionally weird and absolutely require editing by a native English speaker.

Examples are provided in the detailed analysis of the manuscript.

Author Response

Reviewer 2

Comments and Suggestions for Authors

 The article entitled « The human triosephosphate isomerase is a potential target in cancer due to commonly occurring posttranslational modifications » is authored by Sergio Enríquez-Flores, Ignacio De la Mora-De la Mora, Itzhel García-Torres, Luis A. Flores-López, Yoalli Martínez-Pérez and Gabriel López-Velázquez.

This article is a review which proposes an overview of the the possibility to use human triosephosphate isomerase, and in particular its post-translational modifications, as an intervention point in cancer therapy.

The manuscript starts with general information on cancer and its burden on health in modern societies. It then briefly speaks on the techniques available and commonly used for diagnosis, which is followed by another short paragraph on the current options employed to treat patients. The article then dives into the metabolic modifications observed in most cancer cells, and explains how the glycolytic pathway could be used as a weakness to target cancer cells while sparing normal cells. The final part of the manuscript focuses on triosephosphate isomerase, its very particular function in glucose metabolism in cancer cells, and the rationale to turn it into a pharmaceutical target to fight cancer. It then reviews the current knowledge on the enzyme structure, how post-translational modifications alter the structure/function of the enzyme, and how this differs in cancer cells compared to normal cells, thus making the point of using those in cancer therapy.

The review is rather well structured with a classical progression from general data toward very specific points but the organization of the manuscript could be improved as will be proposed below. Some wordings are occasionally weird and absolutely require editing by a native English speaker. The first part of the manuscript seems to have been written by someone less expert than the author of the second part, but homogenizing the work should not take too much effort. The part on metabolism is really well explained, along with the part on HsTPI, which collectively make this review a valuable work.

Response: Thank you for taking the time to review our manuscript. We value your feedback and have made the necessary changes. You can find the updated version of the manuscript with all the changes highlighted in red. We hope you find the updated version to your liking.

Detailed analysis:

The general description of cancer does not provide any original angle and some of the points raised even leave the reader dubious. For example, the claim that “cancer has often been conceptualized wrongly as a disease of old age” (line 54). To make their point, the authors explain that cancer is a significant cause of death in children aged from 1 to 14 (line 57), and that there are more and more new cases of cancers detected in children (line 59). My first objection to this is that it was never said that cancer is “exclusively” a disease of old age. It is instead common knowledge that cancer strikes every age. My second objection is that if the authors are determined to make a point on this, the data they should provide are what part cancer plays in all pathologies in old age, and what part cancer plays in all pathologies in young age, rather than providing data to say that cancer exists in children as well.

Answer: We very much appreciate this observation. Consequently, we have rewritten the first part of the introduction addressing the reviewer's request. Indeed, it is hard to find literature addressing cancer's impact respecting the rest of the pathologies affecting young and old ages. Therefore, we believe the modifications will enrich our manuscript and provide original data regarding cancer. Also, a new figure was included to replace the previous figures 1 and 2 to better illustrate the subject.

Line 68: the authors explain that advancing knowledge on cancer at the molecular scale “has not yet significantly impacted the reduction of new cases”. It is my understanding that curing cancer is far more realistic than aiming to prevent it. So in that respect, the number patients treated every day show that scientific advances deliver what they are expected to, even if better treatments but be found.

Answer: Based on the comments of both reviewers, we have rewritten the first part of the manuscript (previously, lines 30-124).

I would suggest to combine figure 1 and 2 for a comparison at a glance.

Answer: These figures were deleted and replaced by the new Figure 1.

In the paragraph explaining the causes of cancer, I would suggest to use the established term “epigenetic” to encompass the examples provided line 92-93.

Answer: Based on the comments of both reviewers, we rewrote the first part of the manuscript (previously, lines 30-124).

The sentence line 97-98 eludes me. How could risk factors limit cancer development? The sentence should probably be written the other way around.

Answer: Based on the comments of both reviewers, we have rewritten the first part of the manuscript (previously, lines 30-124).

The paragraph dedicated to cancer diagnosis should be improved. It is too brief and does not provide any meaningful addition to the manuscript in its current state. It should at least state that cancer diagnosis relies on biomarkers (may be list a few), and that they may be investigated at the level of DNA or at the level of proteins, with the techniques indicated.

 Answer: The paragraph on cancer diagnosis was integrated into the introduction section, and we did our best to improve it. We took the advice and stated that cancer diagnosis relies on biomarkers. Also, we listed some of them and pointed out that they are investigated at the nucleic acids and proteins level using the indicated techniques.

Same thing on the paragraph speaking about cancer treatment. It is not very meaningful and should be expended. I understand that its purpose is to draw attention to the metabolic differences between cancer cells and normal cells. This sentence should be the last of this paragraph to make the transitions to the next one.

Answer: The paragraph was expanded with new information. We hope it will be more meaningful.

Same thing again on the paragraph on cancer cell metabolism. With the state of the art on this topic, the paragraph could definitely be improved. In its current version, it is too vague. One angle could be a historical perspective? Another angle could be the different types of metabolic changes observed depending on cancer types? But it should definitely be made more specific.

Answer: The paragraph on cancer metabolism was expanded with new and more specific information. We hope it will be sufficiently improved.

Paragraph 5 is where the review really starts for me. Lines 194-195 do not say much about the potential complexity of metabolic rewiring during cancer. I understand that it is not the purpose of this review, but at least the authors should point to one of the many reviews in the field at the end of this sentence.

Answer: Thank you for this observation. Metabolic rewiring was briefly included in this new version.

Same remarks for line 205-206. Provide some examples with accompanying references.

Answer: Thank you for this observation. We provided some examples in this new version.

The paragraph leads to HsTIM, but its deamidation is only mentioned very briefly and I think it might be confusing for lay people to be introduced to a concept that will only be developed in paragraph 7.2.

Answer: We rearrange this idea in the new version.

In paragraph 7.2, the description of deamidation is not correct as some products of the reaction are missing. Deamidation of Asn produces a mixture of Asp and IsoAsp residues, while that of Gln produces a mixture of Glu and IsoGlu.

Answer: You are entirely right. The corrected manuscript includes the correct an extended explanation of the process.

Please explain the meaning of line 302: do the authors mean that deamidation-induced increased turnover of proteins is controversial, or that the outcome of deamidation on protein stability depends on the protein studied? The latter is certainly true in cases were deamidation leads to gain of functions like that of ceruloplasmin in neurodegeneration.

Answer: We want to apologize for this mess in explaining the idea. Actually, we intended to say that deamidation is considered in some studies only as a terminal marking for the HsTIM turnover. In fact, we published years ago a paper using this idea (De la Mora-De la Mora et al., PLoS One. 2015 Apr 17;10(4):e0123379). Now, more data suggest a more complex function of the HsTIM deamidation. Indeed, HsTIM degradation due to deamidation has not yet been demonstrated in vivo. Then, we replaced that part of the text with the following sentence: "Deamidation is considered in some studies only as a terminal marking for the HsTIM turnover. Although the use of deamidation as a molecular timer has been experimentally demonstrated in some proteins (i.e., cytochrome C and aldolase), this is not the case for HsTIM, where this has yet to be probed in vivo."

We hope this new sentence will explain the idea.

Line 306: please modify to “depending on each protein”.

Answer: Done.

Line 320: what is a structural involvement? This is not an established term.

Answer: The correct term is “structural rearrangement.” We changed the text at this point.

Line 336-337 and 442: same remark with the term “HsTIM is highly permeable”. Permeability describes the ability to cross a barrier. May be a better suited word would be accessibility of TIM to small molecules?

Answer: We agree. The sentence was changed to “the deamidated HsTIM highly increases its accessibility to small molecules.”

I am very surprised to find experimental results in a review. I have never seen this before and the format of a review does not transpose to research articles because there is not material and method section. This point is actually what disturbs me most in this whole work. A review article does not contain pieces of research (which would, by the way, be very valuable material for a true research article!).

Answer: We very much appreciate this observation. Although we already published our results about deamidated HsTIM (previously reference 44, now reference 79 in the corrected version), the figures included in this manuscript are unpublished. The figures contain the results of recent assays with methods standardized in our laboratory. Nonetheless, similar figures were already used in a published research article. We preferred to use new pictures rather than take those from the published article. Moreover, we consider it important to show these materials since they are fundamental to support our point of view on studying such post-translational modifications.

On the other hand, molecular models were performed based on the information gathered from the literature but incorporated into in silico representations made in our laboratory.

Based on your request, we stated this in the legend of each figure and added a more detailed description of the methods in the text.

Finally I would recommend to reorganize paragraph 8 and to break it down into one section for each post-translational modification. Also, there should be a paragraph “conclusion” summarizing the manuscript and pointing to future directions.

Answer: We agree and reorganized the paragraph (now paragraph 7 in the corrected version). Also added a Conclusions section.

Minor points:

- modify posttranslational to most-translational in the title and line 20.

Answer: We suppose you meant post-translational. Therefore, posttranslational was modified to post-translational in the title and line 20.

- unclear about the “necessary” line 21.

Answer: We deleted “necessary” from line 21.

- line 31: originating FROM

Answer: That part was rewritten.

- line 32 : weird wording “which can affect humans from mild to very serious”

Answer: That part was rewritten.

- line 33: unclear about “after a period”. What does it mean?

Answer: That part was rewritten.

- line 33: weird wording “divide, and duplicate their number” is redundant.

Answer: That part was rewritten.

- line 43: weird wording “ the WHO has pointed out the preceding”?

Answer: That part was rewritten.

- line 167: energy-demanding

Answer: Done.

- line 168: unclear about the “used to”?

Answer: The complete sentence was rewritten as follows: “The glycolytic pathway rapidly supplies such energetic demand since the speed of the cytosolic ATP generation is approximately 100- times faster than in mitochondria”

- line 226: “both” means that there are 2 subjects. Here there are three: DNA, proteins and lipids.

Answer: We agree. It was corrected.

- line 227: the sentence “covalently…” has no verb. Same in line 300.

Answer: We agree. They were corrected.

- lines 274-278: should be formatted as legend.

Answer: Done.

- line 350: weird wording “whether a cell containing deamidated… could be selectively disturbed”.

Answer: The complete sentence was changed to “A bacterial system lacking endogenous TIM was used to determine the usefulness as a target of the deamidated HsTIM under a cellular environment to obtain selective cytotoxicity.” Also, that paragraph's last part was modified to better explain the idea.

- line 451: referred to AS.

 Answer: Done.

Comments on the Quality of English Language

Some wordings are occasionally weird and absolutely require editing by a native English speaker.

Examples are provided in the detailed analysis of the manuscript.

We received assistance from Dr. Cynthia Fernandez-Lainez in reviewing the latest iteration of our manuscript. She earned her Ph.D. from the University of Groningen in The Netherlands and is currently a PostDoc at the same institution.

Round 2

Reviewer 1 Report

All the queries raised by this reviewer have been attended satisfactorily.

The only point detected is that, among the many new references added to the list, several need corrections:

In references #4, 12, 14, 30, 31, 32, 35, 36, 37, 38, 39, 40, 41, 43, 44, 45, 46, 48, 56, 71, 74, 75, 77, 80, 81, 83 and 128, the name of the journal is not properly abbreviated.

The source of reference #99, which seems to be a preprint, is not identified.

The author of reference #100 is identified as Consortium, T. U., but it should be The UniProt Consortium.

Reviewer 2 Report

I would like to commend the authors for the excellent work they did in modifying their initial manuscript. As a result, the current version now provides a valuable and original view of HsTIM contribution to cancer progression and the possibilities to use it in the future for cancer therapy. Many new citations provide adequate support for the ideas discussed, so again, kudos.

While proofreading this version for final publication, a few very minor mistakes could be corrected:

- line 74 reads awkward. May be a word is missing?

- lines 26, 589, 604, 612 : modify for post-translational

- line 600 : 'thereof' should probably replaced by 'therefore'